# `Plan-SOFAI`: A Neuro-Symbolic Planning Architecture

**Francesco Fabiano,**[1] **Vishal Pallagani,** [2] **Marianna Bergamaschi Ganapini,** [3] **Lior Horesh,** [4] **Andrea Loreggia,** [5] **Keerthiram Murugesan,** [4] **Francesca Rossi,** [4] **Biplav Srivastava** [2]

[1]New Mexico State University,
[2]University of South Carolina,
[3]Union College,
[4]IBM Research,
[5]University of Brescia

ffabiano@nmsu.edu, vishalp@mailbox.sc.edu, bergamam@union.edu, lhoresh@us.ibm.com, andrea.loreggia@unibs.it, keerthiram.murugesan@ibm.com, francesca.rossi2@ibm.com, biplav.s@sc.edu

## Abstract

The notion of Artificial Intelligence (AI) has garnered significant attention in recent years and AI-based tools have increasingly become integrated into our daily lives. As this strand of research is gaining traction, one of the central debates is whether end-to-end Machine Learning or symbolic AI approaches alone can lead to an effective AI model, or if these techniques need to be integrated into a synergistic system. We believe the integration route to be the most promising. To this end, we introduce a specialization of a neuro-symbolic architecture, known as SOFAI (Slow and Fast AI), inspired by the cognitive framework popularized by D. Kahneman's book "Thinking, Fast and Slow". Our system, referred to as `Plan-SOFAI`, aims to tackle planning problems across a large spectrum of scenarios, with a specific focus on the classical setting. `Plan-SOFAI` leverages multiple planning approaches, each possessing distinct characteristics and categorized as either *fast* or *slow* while incorporating a metacognitive process for governance. Finally, we evaluated the performance of this system against state-of-the-art planners, demonstrating that our exhibits a solid balance between solving speed and plans' optimality.

## Introduction

In recent years, the AI community has developed various autonomous and efficient systems and techniques for solving intricate problems across different domains. These problem-solving methods encompass a spectrum, ranging from logic-based approaches to neural-based models. The former focuses on defining rational behavior for AI systems, while the latter seeks to mimic human behavior to some extent by emulating the physiological processes of the human brain.

In this research, we introduce and examine a system that, combining these techniques, addresses planning problems within the *classical* setting. Our work is an extension of the SOFAI architecture (Booch et al. 2021; Ganapini et al. 2022), which, in turn, draws inspiration from the widely recognized cognitive theory *Thinking Fast and Slow* by Kahneman (2011). Our proposed solution incorporates both *fast* and *slow* solvers, along with a metacognitive module responsible for orchestrating their usage. Slow solvers (referred to

as `System-2`) tackle problems through reasoning and symbolic techniques, while fast solvers (`System-1`) rely on past experiences to identify solutions to given problems. The metacognitive module assumes a central role in selecting the most suitable solver for a specific problem.

As previously mentioned, our focus in this paper is on classical planning domains. Our implementation of the SOFAI architecture includes the following components: *i)* existing classical planners to be employed as `System-2` solvers; and *ii)* case-based plans selectors and the so-called *Plansformer* (Pallagani et al. 2023b), as fast solvers. Experimental results conducted on widely recognized planning problem domains demonstrate that our architecture outperforms the use of existing planners alone in terms of striking a balance between solving speed and solution quality.

The paper is structured as follows: after this concise introduction, we provide background knowledge on the "Thinking, Fast and Slow" theory and automated planning. We subsequently outline how we incorporated the core concepts of the thinking fast and slow theory into the planning environment, with a specific focus on the metacognitive module and the solvers. We then present the experimental setup illustrating how our system performs in solving planning problems. We conclude the paper by summarizing our main contributions and offering insights into ongoing work.

## Background

### Thinking Fast and Slow in AI

Thanks to advancements in algorithms, techniques, computational power, and specialized hardware (Marcus 2020), automated reasoning tools have significantly improved in efficiency and reliability within their respective domains. However, all of these tools still lack capabilities that, we humans, naturally associate with the notion of "intelligence" such as generalizability, robustness, and abstraction. Consequently, an increasing part of the AI community is striving to address these limitations by creating systems that exhibit more "human-like qualities". One prominent strategy to address this challenge, adopted by various research groups (Newell 1992; Sun 2006; Goel, Chen, and Wierman 2017; Ganapini et al. 2022), involves the development of tools referred to as *cognitive architectures* (Kotseruba and Tsotsos 2020). These

architectures, usually, aim to combine the strengths of both algorithmic and neural approaches. In particular, in this paper, we explore classical planning in the context of one of these architectures that stems from a modern cognitive theory, *i.e.*, the well-known *Thinking Fast and Slow* paradigm popularized by D. Kahneman in Kahneman (2011).

Kahneman's theory categorizes human reasoning into two distinct `Systems`, denoted as `System-1` (`S1`) and `System-2` (`S2`). In particular, `S1` represents intuitive and imprecise decision-making processes ("thinking fast"), while `S2` provides tools for handling complex decisions through logical and rational thinking ("thinking slow"). Other than problem difficulty, `S1` and `S2` discern which problem they should tackle based on the experience accumulated on the problem itself. That is, when a *new* non-trivial problem has to be solved, it is handled by `S2`. However, some problems initially solvable only by `S2`, can later be solved by `S1` after accumulating sufficient experience. This transition occurs because the procedures employed by `S2` to solve these problems also generate examples that `S1` can later use readily with minimal effort. We note that `S1` and `S2` are not systems in the multi-agent sense, but rather they encapsulate two wide classes of information processing.

## The SOFAI Architecture

As the main contribution of this paper, we present a system inspired by cognitive theories to solve the classical planning problems. In particular, our tool is based on a recent architecture called SOFAI (Booch et al. 2021; Ganapini et al. 2022) that is, in turn, inspired by the dual-system proposed by Kahneman (2011). Following the ideas of Kahneman, the architecture is equipped with two types of `Systems` dedicated to computing a solution to an incoming task, and a third agent in charge of orchestrating the overall reasoning.

In this architecture, incoming problems are initially handled by `S1` solvers that have the required skills to tackle them. `S1` solvers compute a solution relying on the experience collected by the architecture. The computation is not affected by the size of the input problem and thus `S1` solvers provide a solution in constant time.

The solution computed by the `S1` solver (from now on for the sake of simplicity, let us assume it is just one `S1` solver) and the corresponding confidence level is made available to the metacognitive (**MC**) module, presented next, which can now choose between the proposed solution or engaging a `S2` solver. An `S2` agent is typically a reasoning model that is able to deal with the current problem, this kind of solver is more demanding in terms of time and other types of resources. For this reason, only **MC** agent can decide to activate an `S2` solver.

Let us note that, while implementing different strategies, *e.g.*, parallel use of the various systems, is certainly possible, this line of research is focused on investigating the integration of cognitive theory in AI and that is why we are interested in analyzing the 'base case', when only one process can be executed at the time.

**The Metacognitive Module**   Following the SOFAI architecture by Ganapini et al. (2022), let us now present a description of the metacognitive (**MC**) module. SOFAI architecture, and consequently `Plan-SOFAI`, is equipped with a set of mechanisms that allows them to both monitor and control their own cognitive activities, processes, and structures. The goal of this form of control is to improve the quality of the system's decisions. SOFAI has a centralized metacognitive module that exploits both internal and external data and arbitrates between `S1` and `S2` solvers.

This metacognitive module itself follows the *thinking fast and slow* paradigm. This means that **MC** is comprised of two main phases: the first one takes intuitive decisions without considering many factors, while the second one is in charge of carefully selecting the best-solving strategy, considering all the available elements whenever the first phase did not manage to return an adequate solution. We will refer to the former with **MC-1** and to the latter with **MC-2**.

**MC-1** is in charge of deciding whether to accept the solution proposed by the `S1` solver or to activate **MC-2**. **MC-1** takes this decision considering the *confidence*, among other few factors, of the `S1` solver: if the confidence, which usually depends on the amount of experience, is high enough, **MC-1** tries to adopt the `S1` solver's solution.

If **MC-1** decides that the solution of the `S1` solver is not "good enough", it engages **MC-2**. Intuitively, this module needs to evaluate whether to accept the solution proposed by the `S1` solver, use it to guide the `S2` solving procedure, or completely discard it and then activate the `S2` reasoner. To do this, **MC-2** compares the expected reward for the `S2` solver with the expected reward of the `S1` one: if the expected additional reward of running the `S2` solver, compared to the `S1` one, is large enough, then **MC-2** activates the `S2` solver, using the solution proposed by `S1` as heuristic if this is above a certain level of correctness. **MC-2**, following the human reasoning model (Shenhav, Botvinick, and Cohen 2013), is designed to avoid costly reasoning processes unless an even greater expected reward compensates the additional cost for the solution that the `S2` solver will devise.

Let us note that, in this work, to better emulate the expected behavior of a planner we will only allow `Plan-SOFAI` to produce correct solutions.

## Classical Planning

The idea of *automated planning* has been present since the birth of AI and it has been widely explored in the computer science community ever since. This area of research is a branch of Artificial Intelligence where the objective is to find plans, that is, sequences of actions, that can lead some acting agent(s) to achieve desired goals.

Its "basic" form is referred to as *classical planning*. In this setting, in order to have tractable and approachable problems, domains consider constrained environments, *i.e.*, they have to be *i)* static; *ii)* deterministic; and *iii)* fully observable (Ghallab, Nau, and Traverso 2004).

Formally, a classical planning problem can be defined by a tuple $(S, A, I, G)$, where: *i)* $S$ is a finite set of states; *ii)* $A$ is a finite set of actions; *iii)* $I \subseteq S$ represents the initial state; and *iv)* $G \subseteq S$ represents the set of goal states. A solution is then a sequence of actions that transforms the initial

state into a state that satisfies the goal conditions. In classical planning, finding an optimal solution is a key objective, which involves identifying the best possible sequence of actions according to a certain metric, such as the plan's length. Hence, an optimal plan $\pi^*$ from the set of all possible plans $\Pi$ is defined as: $\pi^* = \arg\min |\pi|$ where $\pi \in \Pi$. This optimality criterion ensures that among all the plans that can achieve the goal, the one with the minimum number of actions is selected.

While is not our intention to provide a detailed explanation of the broad field of classical planning, we address the interested readers to Ghallab, Nau, and Traverso (2004); Russell and Norvig (2010); Bolander and Andersen (2011) for a complete introduction on the topic.

## Thinking Fast and Slow in Planning

Two of the prominent lines of work in AI, *i.e.*, data-driven approaches and symbolic reasoning, seem to embody (even if loosely) the two `Systems` presented above. In particular, data-driven approaches share with `S1` the ability to build (possibly imprecise and biased) models from past experience, often represented by sets of data. For example, perception activities, such as seeing, that in humans are handled by `S1`, are currently addressed with Machine Learning (ML) techniques in AI. Similarly, `S2`'s capability to solve complex problems using a knowledge-based approach is somewhat emulated by AI techniques based on logic, search, and planning, which make use of explicit and well-structured knowledge. While the parallelism data-driven–`S1` and symbolic –`S2` represent are a starting point in developing an automated fast and slow AI, we should not assume these two techniques to be the exclusive representative of the respective `System`.

In this paper, we transpose the concepts derived from the thinking fast and slow paradigm into the classical planning setting. We will start by presenting general definitions for `S1` and `S2` solvers and then describe the actual implementations of `S1` and `S2` reasoners in this setting. The general characterization of a `S1` solver, triggered immediately when the problem is presented to `Plan-SOFAI`, does not require many factors. These solvers are assumed to rely on the past experience of `Plan-SOFAI` itself. Moreover, we assume that the running time for `S1` approaches to be independent of the input and, instead, to depend on the experience accumulated by the overall architecture. Finally, we consider a `S1` solver to be an entity that relies on "intuition" (with a slight abuse of notation). Taking into account these characteristics, the next question that naturally arises is *can classical planning ever be considered as `S1` tasks, considering that planners, traditionally, always rely on look-ahead strategies?* We considered some ideas that could help us develop a `S1` planner. Among those, only a few were not using search methods (intensively) but rather mostly relied on experience. Finally, we identified two feasible, yet functional, ways to exploit experience in the aforementioned planning settings.

The first approach makes use of *pre-computed plans*; that is, `S1` can be used to determine which of the plans already generated by past experiences is the one that "fits the best" the current problem. Of course, determining if an already computed plan is a good choice or not for the current problem is a difficult research question on its own. Since the focus of this work is to devise a fast and slow architecture for planning rather than optimizing its internal components, we decided to use a simple, yet effective, criterion to select the best-fitting plan. In particular, the first variation of `S1` selects, among past solutions for the same domain, the pre-computed plan that is the closest in terms of *Levenshtein Distance* and *Jaccard Similarity* (Rinartha, Suryasa, and Kartika 2018), as we will see in more detail later.

Instead, the latter version of `S1`, referred to as Plansformer (Pallagani et al. 2023b), is based on a learning mechanism using LLMs pre-trained in coding languages such as Python, Java, and Ruby. The intuition behind selecting a code-based LLM is to inherit the syntactical knowledge, similar to the family of languages used to define a planning problem. Plansformer is fine-tuned on CodeT5 (Wang et al. 2021) using planning problem instances and their corresponding plans. Given a new problem instance, Plansformer is capable of generating a possibly valid plan, along with the confidence score for the generated plan. The confidence score is computed using the average non-zero probabilities of the generated tokens in the plan returned by Plansformer.

Our `Plan-SOFAI` is a `S1`-by-default architecture: whenever a new problem is presented, a `S1` solver with the necessary skills to solve the problem starts working on it, generating a solution and a confidence level. This allows to minimize the resource consumption making use of the much faster `S1` solving process when there is no need for `S2`—that is when the solution proposed by `S1` is "good enough". Nevertheless, as for the human brain, `S1` may encounter problems that it cannot solve, either due to its lack of experience or the inherent intricacy of the problem itself. These situations require, then, the use of more thought-out resolution processes, generally provided by `S2` approaches. Notice that we do not assume `S2` solvers to be always better than `S1` solvers: given enough experience, some tasks could be better solved by `S1` solvers. This behavior also happens in human reasoning (Gigerenzer and Brighton 2009). Similarly, we don't assume `S1` to be always more efficient than `S2`; in fact, if the problem is very simple it is possible that the "well-thought" process of `S2` could take less time than `S1`.

As for `S2` we considered solving procedures that employ traditional planning strategies. In particular, we used the state-of-the-art *Fast Downward* (Helmert 2006) and *LPG* (Gerevini and Serina 2002) solvers, presented in the background.

## Metacognition Module for Planning

In what follows, we provide a "concrete" view of the `S1`/`S2` framework in the classical planning settings using of Algorithms 1–3.

Before going into detail, let us briefly comment on the input and the parameters of these algorithms. The process requires the domain description (*D*), a particular instance (*I*) that we want to solve on such domain, and the time limit (*TL*) within which the instance needs to be solved. The parameters, instead, represent some internal values that capture some sort of "inclination" of the architecture towards

employing S1. In particular, we have that: *i)* the acceptable correctness (*A*) represents the minimum ratio of solved goals, w.r.t. the total number of them, that defines an acceptable solution. Let us note that this measure can also be changed to depend on other factors or to account for goals' importance, for example. Its default value is 1.0, meaning that Plan-SOFAI will only accept valid plans acting as a full-fledged planner; *ii)* H defines the minimum correctness level (as defined above) to use the solution found by S1 to guide the solving procedure of S2 (if possible). The default value is 0.3; *iii) T1* represents the minimal amount of experience required by Plan-SOFAI to consider a solution proposed by S1. Its default value is set to 20; *iv) T2* represents the minimum number of S1 usages after which it will consider S1 accountable for its mistakes. This threshold allows the architecture to initially try to employ S1 more freely, to augment its experience. Conversely, after the minimum number *T2* of solutions, the metacognition actually uses the previous performances of S1 to check for S1 accountability. Its default value is set to 20; *v) T3* is a value between 0 and 1 and it is used to represent the *risk-aversion* of the architecture: the higher the value the more incline Plan-SOFAI is to use S2. The default value is set to 0.6; *vi)* $\epsilon$ is a factor that is used to scale the probability that S1 solution may actually be employed even if it was not considered convenient. This is added to increase the number of S1 usages, and consequently its experience, in those situations where the low confidence of S1 itself may limit it too aggressively. Let us note that the solution proposed by S1 needs to be validated before being accepted in any case (Line 2 of Algorithm 2). Its default value is 0.1; *vii)* (*M*) represents the experience of Plan-SOFAI. Every time a solution for a problem is found, this is stored in the memory with a series of useful information, *e.g.*, the correctness value, the employed system (*i.e.*, S1 or S2), the difficulty of the instance, the required time, and so on. The default values of these variables were determined through a study aimed at achieving balance between utilizing S1 and S2. We are also currently working on creating automated methods to fine-tune these parameters for optimal resolution of the problem at hand.

We are now ready to describe in more detail Algorithms 1–3. Let us start by presenting Algorithms 1. In particular, we can identify **MC-1** in Lines 1–16, and **MC-2** in Lines 17–39. As already mentioned, to better emulate the thinking fast and slow paradigm, we assume S1 to automatically start and provide a solution at the beginning of **MC-1**. That is why we start the metacognitive process by storing the results of such process in the variables *p* and *cx*, which represent the solution found by S1 and the confidence that S1 has about this solution appropriateness, respectively. The metacognitive process then proceeds to check whether the experience accumulated by the architecture is enough to consider S1 reliable (Line 3). If the architecture has enough experience, the metacognition considers the confidence of S1, adjusted to take into account the previous solutions proposed by S1 itself (Lines 4–12), and determines whether S1's confidence is within the tolerated risk, identified by *T3* (Line 13). If the confidence of S1 is enough, then Plan-SOFAI tries to employ S1's solution (line 14).

---

**Algorithm 1: Fast and Slow Planning Architecture**

**Input**: Domain (*D*), Instance (*I*), Time Limit (*TL*)
**Parameter**: Acceptable Corr. (*A*), *T1, T2, T3*, $\epsilon$, Memory (*M*),
**Output**: Plan (*S*), Correctness (*C*)

1: Let *p* be the solution of *I* found by S1
2: Let *cx* be the confidence of S1 on *p*.
3: **if** $|M.\text{solved\_instances}(D)| \geq T1$ **then**
4:     **if** $|M.\text{solved\_instances}(D, \text{S1})| < T2$ **then**
5:         Let $K = 0$
6:     **else**
7:         Let $avg\_corr = 0$
8:         **for all** $i \in M.\text{solved\_instances}(D, \text{S1})$ **do**
9:             $avg\_corr += \frac{|i.\text{solved\_goals}()|}{|i.\text{tot\_goals}()|}$
10:         **end for**
11:         Let $K = 1 - avg\_corr$
12:     **end if**
13:     **if** $cx \times (1 - K) \geq T3$ **then**
14:         **return** $\langle S, C \rangle = \text{try\_S1}(p, D, I, TL)$
15:     **end if**
16: **end if**
17: Let $diff = I.\text{compute\_difficulty}()$
18: Let $est\_time = M.\text{avg\_time\_from\_diff}(diff)$
19: Let $rem\_time = TL - \text{elapsed\_time}$
20: Let $est\_cost = \frac{est\_time}{rem\_time}$
21: **if** $est\_cost > 1$ **then**
22:     **return** $\langle S, C \rangle = \text{try\_S1}(p, D, I, TL)$
23: **else**
24:     Let $prob = (1 - T3) \times \epsilon$
25:     **if** $prob \geq \text{generate\_random\_number}(0, 1)$ **then**
26:         **return** $\langle S, C \rangle = \text{try\_S1}(p, D, I, TL)$
27:     **else**
28:         $C = \frac{|I.\text{solved\_goals}(p)|}{|I.\text{tot\_goals}()|}$
29:         **if** $C \geq A$ **then**
30:             **if** $(1 - (est\_cost \times (1 - T3))) \geq C \times (1 - K)$ **then**
31:                 **return** $\langle S, C \rangle = \text{S2\_solve}(p, D, I, rem\_time, C)$
32:             **else**
33:                 **return** $\langle S = p, C \rangle$
34:             **end if**
35:         **else**
36:             **return** $\langle S, C \rangle = \text{S2\_solve}(null, D, I, rem\_time, C)$
37:         **end if**
38:     **end if**
39: **end if**

---

If at any point, S1's solution is considered not appropriate by the metacognitive process—because it violates some checks—then **MC-2** starts. This part of the procedure begins by determining a value that represents the difficulty of the problem instance (derived by various factors such as the number of agents, possible actions, fluents, and so on) at Line 17. This measure is then used to determine the average solving time for a given difficulty and to estimate the cost of solving the given problem (Lines 18–20). If the cost exceeds 1 then there is not enough time to call S2 and Plan-SOFAI tries to employ S1. The system can also adopt S1 with a probability that is related to the risk aversion *T3* and a parameter $\epsilon$, this is done in Lines 24–26, to improve the exploration skill of the architecture itself. Plan-SOFAI evaluates the solution proposed by S1 and, if it is acceptable (Line 29), whether the extra time required by S2 counterbalanced the cost (Line 30). If the solution is not acceptable or the in-

crease in correctness is big enough S2 is called (Line 36 and 31, respectively), otherwise the solution proposed by S1 is used (Line 33).

---

**Algorithm 2: try_S1 function**

---

**Input**: Plan ($p$), Domain ($D$), Instance ($I$), Time Limit ($TL$)
**Parameter**: Acceptable Correctness ($A$)
**Output**: Plan ($S$), Correctness ($C$)

1: $C = \frac{|I.\texttt{solved\_goals(p)}|}{|I.\texttt{tot\_goals()}|}$
2: **if** $C \geq A$ **then**
3:     **return** $\langle S = p, C \rangle$
4: **else**
5:     **return** $\langle S, C \rangle$ = S2_solve(*null*,$D$,$I$,$TL$,$C$)
6: **end if**

---

---

**Algorithm 3: S2_solve function**

---

**Input**: Plan ($p$), Domain ($D$), Instance ($I$) Time Limit ($TL$), S1 Correctness ($C$)
**Output**: Plan ($S$)

1: **if** $C \geq H$ **then**
2:     $p\_heur = p$
3: **else**
4:     $p\_heur = \emptyset$
5: **end if**
6: **if** S2 ($D$,$I$, $p\_heur$) terminates within $TL$ **then**
7:     **return** $\langle S = $ S2.get_plan()$, C = 1 \rangle$
8: **else if** $p \: != null$ **then**
9:     **return** $\langle S = p, C = \frac{|I.\texttt{solved\_goals(p)}|}{|I.\texttt{tot\_goals()}|} \rangle$
10: **else**
11:     **OPT-OUT**
12: **end if**

---

Algorithms 2 and 3 are instead used to try and adopt the solution proposed by S1 and to try and solve the problem with S2, respectively. In particular, Algorithms 2 takes the solution proposed by S1 and checks whether it has an acceptable degree of correctness. If it does then the solution is employed, otherwise, Algorithm 3 is called. This function simply calls the S2 approach (*i.e.*, Fast Downward or LPG) on the instance of the problem to solve and, if it terminates before the available time ends, it returns the plan found by the S2 planner with confidence equal to 1. If S2 cannot find the solution within the time limit then the solution from S1, if acceptable, is adopted; otherwise, Plan-SOFAI returns no solution and terminates. Let us note that the initial part of the algorithm (Lines 1–5) checks whether the solution proposed by S1 has a high enough level of correctness to be employed as heuristics for the actual S2 resolution process. If it does, this solution (represented by $p$) is given to S2 so that the planner can make use of this information otherwise, it is discarded and an empty solution is passed to S2. Different S2 solvers might handle partial solutions differently; *e.g.*, in our case, Fast Downward is not able to exploit this extra information while LPG is able to use re-planing techniques on the S1 solution.

## S1 and S2 solvers

In the previous paragraph, we described how our architecture decides which solving approach is the most appropriate, here we will provide a high-level overview of solving processes themselves. In particular, we designed our S1 solvers to solely rely on past experience.

The first type of S1 solver, which is the case-based one, analyzes the memory and looks, through the various solved instances (using either S1 itself or S2), which one is the closest to the problem that is being tackled. Once the closest instance is identified, S1 returns the plan associated with it as a solution and the distance value as a measure for confidence. This distance can be calculated in two different ways, generating effectively two different S1 solvers. The first measure of distance considers the problems as a set of formulae, *i.e.*, the ones that comprise the initial and goal states, and adopts the well-known Jaccard Similarity (Rinartha, Suryasa, and Kartika 2018). The second metric is calculated by transforming the two instances into two distinct strings, once again comprised of all the initial and goal states information (separated by the special characters "|"), that are then compared using the Levenshtein distance (Haldar and Mukhopadhyay 2011) to determine the actual distance measure.

The other type of S1 takes the domain and problem description of a planning instance and maps it to a sequential prompt given as input to Plansformer (Pallagani et al. 2023b). The output obtained is a plan along with the associated confidence score. Plansformer (Pallagani et al. 2023b) is a learning-based planner developed by fine-tuning CodeT5 (Wang et al. 2021), a language model pre-trained on programming languages, using a planning dataset. While there are many Large Language Models (LLMs) to select as a candidate for this work, we shortlist the models pre-trained on code generation to exploit the syntactic information in the programming languages implicitly captured in their weights. CodeT5 is a masked language model consisting of an encoder-decoder stack inspired by the transformer architecture. It is capable of performing a wide range of tasks including code generation and understanding tasks. The CodeT5 model possesses several properties amenable to the planning domain, such as its ability to generate goal-directed, sequential instruction and semantically meaningful program codes with syntactic and structural constraints. The reason for choosing CodeT5 based on empirical evaluation of planning capabilities of different LLMs is presented in (Pallagani et al. 2023a). The planning dataset used to fine-tune CodeT5 encompasses classical planning domains such as **Blocksworld**, **Gripper**, etc. Let us note that this S1 makes use of a pre-computed training set and does not increase its experience during the solving phase. This is a design choice and we leave the exploration of Plansformer with "dynamic" training as a future work. The effectiveness of different S1 systems is highly dependent on the quantity and quality of past plans they utilize. Finally, for the sake of brevity, we refer to (Pallagani et al. 2023a) for a comprehensive examination of how the dataset quality impacts Plansformer's performance and knowledge transferability.

Finally, while for S1 we needed to implement ad-hoc so-

lutions, the same is not true for S2 planners. In fact, as mentioned we employed two state-of-the-art classical planners, *i.e.*, Fast Downward and LPG, as our S2 solvers. Given that explaining how these planners work is beyond the scope of this paper, we can safely assume these approaches to be black boxes that return the best possible solutions if exist. Nonetheless, we refer the interested readers to Helmert (2006); Gerevini and Serina (2002) for a detailed explanation of the internal mechanisms of Fast Downward and LPG, respectively.

Let us just highlight the main differences between the two approaches to better understand in which scenarios one planner could provide a better solving procedure than the other. Classical planning encompasses various strategies for solving complex problems, among which FD and LPG are particularly notable. FD employs a systematic, heuristic-driven search strategy that translates planning problems into multi-valued planning tasks. Utilizing causal graph heuristics, FD is designed to achieve optimal solutions by systematically exploring the search space. This method is well-suited for static environments where domain complexities are predictable and can be encapsulated within a structured search strategy.

In contrast, LPG employs a stochastic local search approach, which is ideal for environments rich in potential solutions. LPG's adaptability is evident in its handling of partial plans, which can be incrementally adjusted in response to changing conditions. This adaptability renders LPG highly effective in dynamic settings that demand frequent plan revisions due to environmental shifts or new data. The strength of LPG in replanning is rooted in its rapid ability to modify partial plans through local search techniques. Such flexibility is vital in non-static environments, enabling LPG to swiftly identify alternate routes to the goal when the initial plan becomes untenable. The local search is fundamental to this process, equipping LPG to traverse a dense search space and pinpoint feasible solutions with minimal computational demand.

## Experimental Results and Discussion

### Experimental Setup

In this section, we compare the new architecture introduced as the main contribution of this paper with Fast Downward (Helmert 2006) and LPG (Gerevini and Serina 2002) that, to the best of our knowledge, are state-of-the-art solvers in the classical planning setting. All the experiments were performed on a 3.00GHz Intel Core i9-13900K machine with 128GB of memory equipped with an NVIDIA GeForce RTX 4090.

To evaluate our architecture we employed 5 well-known classical domains, *i.e.*, **Blocks-World**, **Ferry**, **Gripper**, **Tower of Hanoi**, and **Miconic**. In the interest of brevity, we will not introduce the domains and we will direct the interested readers to (Seipp, Torralba, and Hoffmann 2022).

The experiments are comprised of 500 different problems, 100 instances for each domain, that vary different characteristics of the domain, the initial state, and the goal state. Every instance is automatically generated, and this generation process is random. Subsequently, the optimal plan length of each instance is determined by solving the problem using Fast Downward. The optimality of a solution is assessed by calculating the extra length in comparison to the optimal plan, expressed as a percentage. If the optimal plan has a length of $o$ and the calculated plan has a length of $l$, then the optimality is computed as $((l-o)/o) * 100$. In this context, the ideal optimality value is represented by "+0.0%".

Regarding the various input and parameters of the architecture (used in Algorithms 1, 2, 3) we imposed: *i)* a Time Limit (*TL*) of 60s to solve each instance; *ii)* an Acceptable Correctness (*A*) of 1.0, meaning that all the goals must be satisfied for a S1 solution to be considered; *iii)* the various thresholds (*i.e.*, *H*, *T1*, *T2*, *T3*) and $\epsilon$ to have their default values; and *iv)* the Memory (*M*) to be initially filled with 25 solved problems (5 from each domain).

### System-1 Selection

The first set of experiments of our works is used to identify the "best" S1 approach to reduce the scope of in-depth analyses required to evaluate Plan-SOFAI against the baseline. These results are presented in Table 1. Let us note that, following each attempted resolution by S1, if the problem remains unsolved, an instance of a S2 solver (*i.e.*, Fast Downward) is invoked to increase the available experience of the S1 solver itself.

To streamline the discussion, we have designated the various S1 with the following abbreviations:

- RNG: this approach randomly selects one of the instances in memory as a potential solution. This method serves as a baseline for comparison, aiming to be outperformed by a more strategically configured S1.
- LEV: in this approach, the solver picks as a possible solution the plan of the instance that minimizes the Levenshtein distance w.r.t. the problem at hand;
- JAC: similarly to the LEV, this approach makes use of the Jaccard distance to select a possible plan;
- CB: this approach combines both LEV and JAC and selects the solution proposed by the method with the highest confidence;
- PF: this approach employs Plansformer as solver;
- MIX: this approach entails selecting the solution with the highest (normalized) confidence score from among all available S1 methods described above.

It is evident from the information presented in Table 1 that PF outperforms the others in terms of accuracy. This method not only produces the highest number of successfully solved and optimal plans but also delivers the best results concerning optimality. The primary drawback of utilizing a Plansformer-based approach is the relatively longer inference time, approximately 2 seconds, much higher than other methods. Although a 2-seconds delay may not be considered a significant overhead for more challenging instances, it could pose an issue when dealing with easy planning instances that could be resolved in mere milliseconds by a S2 solver. Considering that the primary objective of this study is to introduce an architecture capable of effectively combining

| | RNG | LEV | JAC | CB | PF | MIX |
|---|---|---|---|---|---|---|
| Valid Plans | 9 (1.80%) | 117 (23.40%) | 79 (15.80%) | 117 (23.40%) | **402 (80.40%)** | 395 (79.00%) |
| Optimal Plans | 0 (0.00%) | 54 (10.80%) | 42 (8.40%) | 54 (10.80%) | **386 (77.20%)** | 379 (75.80%) |
| Time (avg) | **0.002s** | **0.003s** | **0.002s** | **0.003s** | 2.079s | 2.057s |
| Correctness (avg) | 0.106 | 0.512 | 0.483 | 0.511 | **0.943** | 0.921 |
| Optimality (avg) | +24.15% | +5.56% | +5.56% | +5.56% | **+0.34%** | +0.42% |

Table 1: Comparison between the various `S1` solvers treated as standalone planners. The percentage numbers of correct and optimal plans (Lines 2 and 3 of the Table, respectively) are relative to the whole batch of experiments, *i.e.*, 500 instances.

`S1` and `S2` planning approaches, rather than delving into the details of these approaches individually, we will postpone the investigation of Plansformer optimization to future research. Nevertheless, we will regard `PF` as the most promising approach at our disposal and will employ this configuration as the `S1` component of `Plan-SOFAI` for the forthcoming set of experiments. Let us note that we tested all the possible configurations (not reported to avoid unnecessary clutter) and the results reflect what was expected, *i.e.*, using `Plan-SOFAI` with the best `S1` approach produces the best results.

## Experimental Results

The experiments conducted in this section are intended to determine whether `Plan-SOFAI` can be considered an effective substitute for state-of-the-art planners. It is worth noting that while `Plan-SOFAI` can be utilized to accept partial solutions, depending on the value of the Acceptable Correctness parameter (*A*), thereby offering a valuable tool for resource-constrained scenarios, in this particular context, we exclusively evaluate `Plan-SOFAI` as a fully-fledged planner. Evaluations of `Plan-SOFAI` with partial solutions can be found in (Fabiano et al. 2023)

As a baseline for our experiments, we used the solvers Fast Downward (FD) (Helmert 2006) and LPG (Gerevini and Serina 2002). We let the solvers tackle all the instances with a time-out of 60 seconds per instance. The main idea is that these approaches represent the current capabilities of classical planning and are comparable to a solely `S2`-based architecture.

We then compared the baseline results with different configurations of the architecture outlined in this paper (Table 2). To avoid unnecessary clutter, let us identify the various configurations, with the following abbreviations:

- `FD`: This configuration represents the Fast Downward planner with the heuristics `lmstar(blind())`. It aims to find solutions while also seeking optimality and is our baseline for solvers that do not heavily rely on local search and, consequently, produce reasonably sized solutions.
- `LPG`: This setting we employs the LPG planner with the `-quality` modality. The approach is geared toward finding solutions using local search and striving for maximum optimality. Since LPG incorporates local search, the achieved solution optimality tends to be relatively low.
- `SOFAI-PF-FD`: In this setup, `S1` is identified by `PF` (as above) and `S2` is `FD` (as described in the first item of this list). If the solution returned by `PF` does not have a correctness $\geq A$, *i.e.*, in our case $\geq 1.0$, `S2` is adopted.

- `SOFAI-PF-LPG`: This configuration is similar to `SOFAI-PF-FD` but utilizes `LPG` as `S2` instead of `FD`.
- `SOFAI-PF-FDxLPG`: In this configuration, `PF` represents the `S1`, and `S2` is a combination of `FD` and `LPG`. In particular, `LPG` serves as a planner capable of adapting partially complete plans (in this case generated by `PF`) into complete plans. If the correctness level of the plan from `PF` is $\geq A$, that plan is directly output. However, if the correctness level is higher than *H* (but still below *A*), the partially correct plan from `PF` is adapted using `LPG`. Finally, if the correctness of `S1` is lower than *H*, `FD` is used to solve the problem from scratch.
- `SOFAI-PF-LPGxLPG`: This configuration is similar to `SOFAI-PF-FDxLPG` but utilizes `LPG` as a `S2` instead of `FD` when `Plan-SOFAI` needs to replan from scratch, *i.e.*, when the solution proposed by `PF` has correctness $< H$.

## Discussion

Table 2 provides some noteworthy findings. Let us begin our examination by focusing on our baseline, which encompasses `FD` and `LPG`. Notably, it is evident that although `FD` displays the most favorable optimality metrics, it falls short in resolving specific issues, resulting in higher overall resource consumption. Conversely, `LPG`, thanks to its local-search-based approach, manages to solve all the problems with a low time consumption, rendering it the most efficient strategy in this regard. However, it is worth mentioning that the use of local search often leads to `LPG` producing solutions with less-than-ideal optimality statistics. Subsequently, `PF` alone, which represents the best `S1` planner at our disposal, resolves a substantially smaller subset of issues in comparison to both `FD` and `LPG`. Nevertheless, it demonstrates a robust optimality metric, and an acceptable average planning time (thanks to its constant-time solving process).

Evidently, an ideal outcome would involve combining the optimality characteristics of `FD` with the fast solving time of `LPG`. Unfortunately, such a synergy cannot be achieved as the distinct problem-solving processes are seen as black-boxes to one another and combining them in a unique tool would necessitate a substantial effort comparable to devising an entirely new solver. In this context, our proposed system, `Plan-SOFAI`, offers an alternative for enabling diverse solvers to interface and collaborate.

The adaptable architecture `Plan-SOFAI` allows us to merge the quick problem-solving capacity of `PF`, albeit less precise than the baseline, with the baseline itself in various ways. This enables the development of an approach that simultaneously maintains strong optimality and augments the

| | FD | LPG | PF | SOFAI-PF-FD | SOFAI-PF-LPG | SOFAI-PF-FDxLPG | SOFAI-PF-LPGxLPG |
|---|---|---|---|---|---|---|---|
| Valid Plans | 451 (90.20%) | **500 (100.00%)** | 402 (80.40%) | 483 (96.60%) | **500 (100.00%)** | 490 (98.00%) | 490 (98.00%) |
| Optimal Plans | 451 (90.20%) | 264 (52.80%) | 386 (77.20%) | **464 (92.80%)** | 434 (86.80%) | 445 (89.00%) | 448 (89.60%) |
| S1 Plans | – | – | 402 (80.40%) | 398 (79.60%) | 401 (80.20%) | 397 (79.40%) | 394 (78.80%) |
| Optimal | – | – | 386 (96.02%) | 379 (95.23%) | 383 (95.51%) | 377 (94.96%) | 378 (95.94%) |
| S2 Plans | 451 (90.20%) | 500 (100.00%) | – | 85 (17.00%) | 99 (19.80%) | 93 (18.60%) | 96 (19.20%) |
| Optimal | 451 (100.00%) | 264 (52.80%) | – | 85 (100.00%) | 51 (51.52%) | 68 (73.12%) | 70 (72.92%) |
| FD | 451 (90.20%) | – | – | 85 (17.00%) | – | 5 (1.00%) | – |
| Optimal | 451 (100.00%) | – | – | 85 (100.00%) | – | 5 (100.00%) | – |
| LPG | – | 500 (100.00%) | – | – | 99 (19.80%) | – | 5 (1.00%) |
| Optimal | – | 264 (52.80%) | – | – | 51 (51.52%) | – | 2 (40.00%) |
| PF + LPG | – | – | – | – | – | 88 (17.60%) | 91 (18.20%) |
| Optimal | – | – | – | – | – | 63 (71.59%) | 68 (74.73%) |
| Time (avg) | 8.479s | **0.675s** | 2.079s | 4.915s | 2.318s | 2.199s | 2.163s |
| Optimality (avg) | **+0.00%** | +23.68% | +0.34% | +0.02% | +9.78% | +1.13% | +4.86% |

Table 2: Comparison of the various configuration of `Plan-SOFAI` with `FD` and `LPG`. Each method is divided into its internal processes, highlighting the contribution of each solving tool. The gray lines indicate the percentage of plans relative to the line above. Line 12 presents statistics for "`PF + LPG`" indicating the number of times a not-valid plan found by `PF` was considered 'good enough' (correctness $\geq H$) to be used as a partial solution for a subsequent LPG replanning.

problem-solving spectrum (w.r.t. `PF`), consequently reducing execution time. The best example of this approach is `SOFAI-PF-FDxLPG`, which, in our estimation, represents the most balanced trade-off among all the analyzed planning techniques. This instance takes full advantage of the solutions generated by `S1` and of the two `S2` solvers. Although the solving time is highly dependent on the inference time of `PF`, any enhancement of this problem-solving technique could yield an even more rapid architecture. We leave the optimization of `PF`, and the investigation of new `S1` solvers for future research.

Let us emphasize that this instance of `Plan-SOFAI`, while rooted in specific `S1` and `S2` solvers, embodies a broader contribution. We introduce a methodology for merging diverse planning techniques within a unified architecture, mirroring human cognition by capitalizing on both symbolic and experience-based reasoning.

Furthermore, the flexibility of `Plan-SOFAI` extends to modifications in `S1` and `S2` solvers, potentially yielding numerous `Plan-SOFAI` instances tailored to distinct scenarios. In this regard, `Plan-SOFAI` serves as a centralized tool for maximizing the variety of planners found in the literature, combining them with other techniques to address their respective limitations. This adaptability also extends to internal parameters, permitting users to fine-tune the architecture for accommodating less accurate solutions in the interest of time savings or vice versa. For example, if we wanted to `Plan-SOFAI` to domains characterized by low solution density, it is reasonable to forego reliance on `LPG` and instead explore alternative strategies for harnessing precomputed plans, such as leveraging `PF` to compute plan prefixes and reduce the computational burden on `FD`.

Ultimately, `Plan-SOFAI` is not confined to the classical planning scenarios. Thanks to its capacity to adapt to resource constraints, this architecture can be useful in resolving problems within settings where inherent complexity, *e.g.*, multi-agent epistemic planning (Bolander 2017), would otherwise render the planning process infeasible. This is illustrated in Fabiano et al. (2023) where an older version of the architecture, instantiated for multi-agent epistemic planning, is presented. In such intricate cases, the trade-off of accepting partially correct but sound plans becomes a viable means of producing solutions that, albeit partial, possess useful information.

## Conclusions and Ongoing Work

In this work, we presented an architecture to tackle planning problems, in classical settings, that is heavily inspired by the well-known cognitive theory Thinking Fast and Slow by Kahneman (2011). This tool builds on the SOFAI architecture (Ganapini et al. 2022), which makes use of a metacognitive process to arbitrate the solving processes, and two solvers referred to as `S1` and `S2`. While `S2` is directly derived from the literature the `S1` solvers have been designed ad-hoc for the proposed architecture to exploit past experience. The SOFAI-inspired approach showed very promising results demonstrating the ability to combine diverse solving approaches to enhance their performances. Another advantage of the proposed architecture is that it can be used to incorporate new solving techniques developed by the community. In fact, our tool can be easily modified to employ different `S1` or `S2`, or multiple versions of them.

While the obtained results are promising, we are continuously working on improving `Plan-SOFAI`. First, we are devising ways of making our architecture easy to use with different solving approaches. That is, we are working on an encoding of `Plan-SOFAI` that is completely independent of the solving approach where the interested researcher can plug multiple planners, as `S1` or `S2` solvers, to test their interaction. To this end, we are also working on devising new strategies to incorporate the solutions proposed by `S1` into the `S2` solving process, independently from the latter while also automatically adjusting the default values of the internal thresholds. An example of such strategies would be to let `S1` generate partial solutions that can create new initial states, where a set of sub-goals is already satisfied, to reduce the workload of the `S2` solvers. Finally, we are also working on devising and optimizing existing experience-based planners. Most notably we are investigating ways of integrating the concept of *continual learning* in `PF`, allowing this technique to learn over time when employed in `Plan-SOFAI`.

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
