# OpenReview forum: "Plan-SOFAI: A Neuro-Symbolic Planning Architecture"
_AAAI.org/2024/Workshop/NuCLeaR — NuCLeaR 2024_

### Official Review · Reviewer_uUbg · 2023-12-07
**Plan-SOFAI: A Neuro-Symbolic Planning Architecture**

**Rating:** 5
**Confidence:** 5

**Review:**

The paper presents a novel planner that merges a data-driven planner, such as Plansformer (referred to as System 1 or fast thinking), with a traditional planner like LPG or FD (termed System 2 or slow thinking). The concept behind this approach is to employ System 1 to generate a plan, while a metacognitive system is employed to assess whether to go ahead with the plan based on the confidence score of this plan. When the confidence score is low, the system resorts to utilizing System 2 to formulate a plan for the given problem.

Questions: How are the initial values for the parameters H, T1, T2, T3, and epsilon determined? Are they subject to any learning process?

Could you please explain the reason behind PF, a fast-thinking approach, showing slower execution times than LPG as indicated in Table 2?

Given the challenges of representing intricate domain knowledge using System S2 in practical planning situations, isn't it better to adopt approaches where System S1 (based on past experiences) and System S2 (based on available incomplete knowledge) work together as in Rana et al. in their paper "Sayplan: Grounding large language models using 3D scene graphs for scalable task planning" (arXiv preprint arXiv:2307.06135, 2023)? Is the approach of trying S1 and then resorting to S2 in case S1 fails feasible as S2 itself may not be available for the domain.

---

### Official Review · Reviewer_Te8Y · 2023-12-07
**The paper shows promise but can benefit from more ablations and clear motivating example.**

**Rating:** 6
**Confidence:** 3

**Review:**

Strengths:
Combines approach of think fast and slow.  The results show that the technique works.

Weaknesses:
1. Lack of motivating examples to support the proposed approach. I don't understand why the authors are first doing neural and logical reasoning in a fallback manner, there can also be a case that they could be executed parallelly - a voting based technique combining the approaches (in cases where the symbolic latency is less) can work better?

Formatting:
1. It is difficult to parse and understand the table
2. It would be much easier to split it into two and have the techniques as rows and metrics as columns.

---

### Official Review · Reviewer_ws2V · 2023-12-08
**This review assesses the Plan-SOFAI NeSy planning architecture, highlighting its effective integration of diverse AI techniques and its potential for addressing real-world planning problems. While the paper shows promise, there is room for further optimization and testing across a broader range of scenarios.**

**Rating:** 7
**Confidence:** 3

**Review:**

The paper proposes a specialized NeSy architecture based on the SOFAI model, inspired by Kahneman's "Thinking, Fast and Slow" paradigm. This architecture, referred to as Plan-SOFAI (Slow and Fast AI), aims to address various planning problems across multiple scenarios by leveraging both fast and slow planning approaches while incorporating a metacognitive process for governance.

The paper is well-written and easy to follow, with clear technical explanations. The authors have demonstrated a seamless integration of different AI techniques (symbolic and end-to-end Machine Learning) within a single architecture, including its risk aversion parameters and the process for adopting solutions proposed by System-1 (S1) and System-2 (S2) solvers. The straightforward integration of the system shows promise in addressing real-world planning problems. The proposed architecture is compared with state-of-the-art planners, displaying a good balance in terms of solving plan optimality, which is crucial for practical applications.

However, the paper mentions postponing the investigation of the optimization of the Plansformer, a key component of System-1. Although this could limit the current effectiveness of the system, the authors have expressed their interest in further improvements and optimizations as the next step in future work. Furthermore, while the paper evaluates the architecture against established planners, the testing scenarios and domains may not fully capture the diverse range of real-world applications and challenges. For instance, it is not clear from the experiments whether the architecture's effectiveness is constrained by the quality and quantity of past solved problems used in its memory, especially in addressing new or significantly different scenarios.

In summary, Plan-SOFAI, as a NeSy Planning Architecture, contributes to AI research by effectively integrating diverse planning techniques. While it requires further enhancement in optimization and testing across diverse scenarios, I believe the paper qualifies for presentation at the NuCLeaR workshop.

---

### Decision · Program_Chairs · 2023-12-11

Accept